# User-Friendly Reverse Genetics System for Modification of the Right End of Fowl Adenovirus 4 Genome

**DOI:** 10.3390/v12030301

**Published:** 2020-03-11

**Authors:** Bingyu Yan, Xiaohui Zou, Xinglong Liu, Jiaming Zhao, Wenfeng Zhang, Xiaojuan Guo, Min Wang, Yingtao Lv, Zhuozhuang Lu

**Affiliations:** 1College of Marine Science and Biological Engineering, Qingdao University of Science and Technology, Qingdao 266042, China; yanbingyusd@163.com (B.Y.); lxlsddz01@163.com (X.L.); 2State Key Laboratory of Infectious Disease Prevention and Control, National Institute for Viral Disease Control and Prevention, Chinese Center for Disease Control and Prevention, Beijing 100052, China; zouxh@ivdc.chinacdc.cn (X.Z.); zhaojm777@163.com (J.Z.); zhangwfme@163.com (W.Z.); guoxj@ivdc.chinacdc.cn (X.G.); wangmin@ivdc.chinacdc.cn (M.W.); 3Department of Laboratory Medicine, School of Public Health and Management, Weifang Medical University, Weifang 261053, China; 4Center for Biosafety Mega-Science, Chinese Academy of Sciences, Wuhan 430071, China; 5Chinese Center for Disease Control and Prevention-Wuhan Institute of Virology, Chinese Academy of Sciences Joint Research Center for Emerging Infectious Diseases and Biosafety, Wuhan 430071, China

**Keywords:** fowl adenovirus, reverse genetics system, Gibson assembly, restriction enzyme, essential gene

## Abstract

A novel fowl adenovirus 4 (FAdV-4) has caused significant economic losses to the poultry industry in China since 2015. We established an easy-to-use reverse genetics system for modification of the whole right and partial left ends of the novel FAdV-4 genome, which worked through cell-free reactions of restriction digestion and Gibson assembly. Three recombinant viruses were constructed to test the assumption that species-specific viral genes of ORF4 and ORF19A might be responsible for the enhanced virulence: viral genes of ORF1, ORF1b and ORF2 were replaced with GFP to generate FAdV4-GFP, ORF4 was replaced with mCherry in FAdV4-GFP to generate FAdV4-GX4C, and ORF19A was deleted in FAdV4-GFP to generate FAdV4-CX19A. Deletion of ORF4 made FAdV4-GX4C form smaller plaques while ORF19A deletion made FAdV4-CX19A form larger ones on chicken LMH cells. Coding sequence (CDS) replacement with reporter mCherry demonstrated that ORF4 had a weak promoter. Survival analysis showed that FAdV4-CX19A-infected chicken embryos survived one more day than FAdV4-GFP- or FAdV4-GX4C-infected ones. The results illustrated that ORF4 and ORF19A were non-essential genes for FAdV-4 replication although deletion of either gene influenced virus growth. This work would help function study of genes on the right end of FAdV-4 genome and facilitate development of attenuated vaccines.

## 1. Introduction

Fowl adenoviruses (FAdV) belong to the family of Adenoviridae and the genus of *Aviadenovirus* [1]. Currently 12 types of FAdV have been identified (FAdV-1 to -8a and -8b to 11), which are further classified into five species (*FAdV-A to -E*) [2]. FAdVs are believed to be the pathogens causing adenoviral gizzard erosion (AGE), inclusion body hepatitis (IBH) and hepatitis-hydropericardium syndrome (HHS) in chickens, although the outcome of anFAdV infection depends on many factors including serotype and strain of the virus, type and age of the host, and the route of inoculation [3]. FAdV-4 is the predominant causative agent of HHS, which is characterized by an accumulation of clear, straw-colored fluid in the pericardial sac and a swollen, friable, discolored liver with focal necroses and petechial hemorrhages. The first outbreak of FAdV-4 occurred near Karachi, Pakistan in 1987 [4]. After that, the virus has spread to most of Asia, Central and South America, and some European countries. Since July 2015, outbreaks of HHS caused by a novel genotype of FAdV-4 have been reported in China, causing significant economic losses to the poultry industry [5,6,7].

The family of Adenoviridae is composed of five genera, among which the Mastadenovirus, especially human adenovirus species C (HAdV-C), is extensively studied. The adenoviral genes are artificially classified into two groups: the genus-common genes, which are located in the central genome and reserved in all genera, encode structure, replication and encapsidation proteins; and genus-specific genes, which are found at both ends of the viral genome, encode non-structural proteins and play important roles in virus–host interaction [2,8]. Generally, genus-common genes are essential for virus replication in vivo or in vitro cell culture. The function of these genes should be conserved among the family and then be analogous to that of Mastadenovirus. Although most of the genus-specific genes in HAdV-C have been studied, very few genus-specific genes in FAdV have been elucidated in their functions. The reverse genetics system is a valuable approach to study viral genes of FAdV, and some systems have been established and used to construct gene transfer vectors and vectored vaccines [9,10]. Most of these systems work through homologous recombination, which needs the activity of intracellar recombinase.

Here, we attempted to establish a reverse genetics system for the novel strain of FAdV-4 isolated from recent outbreak in China, which was different from existing ones and could be easily applied with our recently introduced technique of restriction-assembly [11,12]. We also characterized some genus-specific genes of FAdV-4 by using this system.

## 2. Materials and Methods

### 2.1. Cells, Plasmids and Oligonucleotides

Chicken hepatoma cells (Leghorn Male Hepatoma, LMH) were purchased from American Type Culture Collection(ATCC CRL-2117, Manassas, VA, USA) and were maintained in Dulbecco’s modified Eagle’s medium (DMEM) plus 10% fetal bovine serum (FBS; HyClone, Logan, UT, USA) at 37 °C in a humidified atmosphere supplemented with 5% CO_2_, and passaged twice a week. Flasks or plates were precoated with 0.1% gelatin to help LMH cells spread evenly according to the instructions of ATCC. Plasmid pKFAV4 (plasmid bearing Kanamycin-resistant gene and FAdV-4 genome) was constructed in the laboratory previously [13], which contained the complete genome of the novel FAdV-4 (Genbank accession No. MG547384). pAd5GFP was an E1/E3-deleted HAdV-5-based adenoviral plasmid constructed previously in the laboratory [14], which contained an Enhanced Green Fluorescent Protein (GFP) expression cassette controlled by the human cytomegalovirus (CMV) promoter and SV40 polyA signal.

Plasmid transformation was performed on *Escherichia coli* TOP10 chemically competent cells with the standard heat shock procedure according to the manufacturer’s instructions (TIANGEN Biotech, Beijing, China).

For virus rescue or amplification, confluent LMH cells were split at a ratio of 1:3. When the cells had reached 90% confluence, transfection or infection was carried out. After incubation for indicated time, the culture medium was replaced with fresh DMEM containing 2% FBS. 

Single-stranded DNA oligos were chemically synthesized. They were used as the primers for Polymerase Chain Reaction (PCR) or used to assemble short double stranded DNA through overlap extension PCR. Information related to these oligos was summarized in Table 1.

### 2.2. Construct Intermediate Plasmid pKFAV4AP

Plasmid pKFAV4 (46.2 kb) contained two AvrII sites. Two single-stranded oligonucleotides surrounding the AvrII sites in pKFAV4 were synthesized (Table 1) and fused to form a double stranded DNA (AvrII-PacI, 96 bp in length) through DNA polymerasemediated extension reaction (Q5 High-Fidelity DNA Polymerase, Cat. M0491S; New England Biolabs, Ipswich, Massachusetts, USA), and a PacI site (ttaattaa) was added in the middle to demarcate the two AvrII-centered parts. pKFAV4 was digested with AvrII, the small fragment (16.7 kb) was recovered from agarose gel (Zymoclean Large Fragment DNA Recovery Kit, Cat.D4045; Zymo Research, Irvine, CA, USA) and mixed with AvrII-PacI fragment, and Gibson assembly was performed to generate an intermediate plasmid pKFAV4AP (pKFAV4
AvrII-PacI) (NEBuilderHiFi DNA Assembly Master Mix, Cat. E2621; New England Biolabs).

### 2.3. Delete ORF1-ORF2 in pKFAV4

pKFAV4AP was digested with AgeI/NheI, and the fragment of 4942 bp was recovered; PCR was performed to amplify AgeI-ORF1 fragment (Open Reading Frame, ORF; from upstream of AgeI site in ORF0 to ORF1; 204 bp) with primers of 1707KFAV4AgeIF/R (Table 1) using pKFAV4AP as the template (Q5 High-Fidelity DNA Polymerase); PCR was performed to amplify GFP expression cassette (1672 bp) with primers of 1707F02GFPF/R using pAd5GFP as the template; and these three fragments were fused together to generate plasmid pKFAV4APN-GFP (pKFAV4AP
NheI-GFP) by DNA assembly. pKFAV4APN-GFP was digested with NheI, treated with CIP (Calf Intestinal Alkaline Phosphatase; New England Biolabs), and ligated with the large fragment of NheI-digested pKFAV4AP (9840 bp) to generate plasmid pKFAV4AP-GFP (16,575 bp). pKFAV4AP-GFP was digested with PacI and fused to the large fragment of AvrII-digested pKFAV4 (29,468 bp) to generate pKFAV4-GFP (45,981 bp) by DNA assembly (NEBuilderHiFi DNA Assembly Master Mix).

### 2.4. Replace Coding Sequence of FAdV-4 ORF4 with That of mCherry in pKFAV4-GFP

PvuI and BamHI were single cutters flanking ORF4 coding sequence (CDS) in pKFAV4APN-GFP. A fragment from the PvuI site to the start codon of ORF4, mCherry CDS and a fragment from the stop codon of ORF4 to the BamHI site were amplified by PCR, respectively (primers 1711FAV4GCX5-10, Table 1); and these three fragments were fused together by overlap extension PCR (1281 bp). The 1281-bp fragment was digested with PvuI/BamHI and used to replace the corresponding fragment in pKFAV4APN-GFP to generate pKFAV4APN-GX4C (pKFAV4APN
GFP-xORF4-mCherry, 6948 bp) through restriction-ligation cloning. pKFAV4APN-GX4C was digested with NheI, treated with CIP and restored to pKFAV4AP to generate pKFAV4AP-GX4C through restriction-ligation cloning. pKFAV4AP-GX4C was digested with PacI and fused to the large fragment of AvrII-digested pKFAV4 to generate pKFAV4-GX4C (46,194 bp) through DNA assembly.

### 2.5. Remove the AvrII Site in Fiber-2 Gene in pKFAV4 and Reduce the Size of the Intermediate Plasmid

PCR was performed to amplify a 117-bp linker with primers of 1812FAV4APf and 1812FAV4APxAr using pKFAV4AP as the template. pKFAV4AP was digested with AvrII to recovered the 16,728 bp fragment, which was fused to the 117-bp PCR product to generate plasmid pKFAV4APX (pKFAV4APxAvrII) through DNA assembly. pKFAV4APX was digested with PacI and fused to the large fragment of AvrII-digested pKFAV4 to generate pKFAV4M (pKFAV4AvrII site-Mutated, 46,196 bp) through DNA assembly.

Overlap extension PCR was employed to generate a 116-bp linker by mixing four primers of 1812FAV4SAP1-4 (Table 1). pKFAV4AP-GFP was digested with AvrII/SpeI, and the fragment of 9476 bp in length was recovered from agarose gel and fused to the 116-bp fragment to generate pKFAV4SAP-GFP (pKFAV4
SpeI-AvrII-PacI GFP, 9540 bp) through DNA assembly.

### 2.6. Delete ORF19A in pKFAV4-GFP

PCR was performed to amplify an 804 bp fragment with primers of 1902FAdV4Xorf19a1/2 using pKFAV4AP as the template, which was digested with SpeI/BstZ17I and used to replace the corresponding region in pKFAV4SAP-GFP to generate pKFAV4SAPX19a-GFP (pKFAV4SAP
xORF19A
GFP, 7119 bp) through restriction-ligation cloning. GFP was further replaced with mCherry gene in pKFAV4SAPX19a-GFP. Part of the CMV promoter (395 bp) was amplified with primers of 1904FAV4MCHE1/2 using pKFAV4-GFP as the template; mCherry CDS (733 bp) was amplified with primers of 1904FAV4MCHE3/4 using pmCherry-N1 (Clontech Laboratories, Mountain View, CA, USA) as the template; and these two PCR products were fused to generate a 1102 bp DNA fragment through overlap extension PCR. The PCR product was digested with NdeI/SalI and used to replace the corresponding region in pKFAV4SAPX19a-GFP to generate pKFAV7807-Che (7807 bp) through restriction-ligation cloning. pKFAV7807-Che was linearized with PacI and fused to the large fragment of SpeI/AvrII-digested pKFAV4M to generate pKFAV4-CX19A (pKFAV4 mCherry xORF19A, 43,528 bp) through DNA assembly.

### 2.7. Rescue, Amplification, Purification and Titration of Recombinant Virus

Adenoviral plasmid (pKFAV4-GFP, pKFAV4-GX4C or pKFAV4-CX19A) was linearized by PmeI-digestion, recovered by ethanol precipitation and used to transfect LMH cells after mixing with Lipofectamine 3000 reagents (Thermo Fisher Scientific, Waltham, MA, USA). Rescued viruses were amplified in LMH cells. Recombinant viruses were purified with standard CsCl discontinuous density gradient ultracentrifugation except that 10 mM citrate (pH6.2) instead of 10 mM Tris-Cl (Ph7.6) was used as the buffer medium for preparing the CsCl solutions [13]. Adenoviral band was collected, dialyzed against a solution containing 10 mM sodium citrate, 150 mM sodium chloride and 5% glycerol (pH6.2), aliquoted and frozen at −80 °C. Particle titer (viral particles per mililiter, vp/mL) of purified virus was determined by measuring the content of genomic DNA where 100 ng of genomic DNA is equivalent to 2.3 × 10^9^ viral particles, since a 43-kb genome has a molecular mass of 2.6 × 10^7^. Infectious titer (infectious units per mililiter, IU/mL)was determined on LMH cells with the limiting dilution assay by counting GFP+ or mCherry+ cells two days after infection [15].

### 2.8. Identification of Recombinant Virus

Virus genomic DNA was extracted from virus-infected LMH cells using a modified Hirt’s method [11,16]. Viral DNA was digested with restriction enzymes and resolved on 0.6% agarose gel containing ethidium bromide by electrophoresis. Primers were designed, synthesized and used for identification of viral genome by PCR (Table 1).

### 2.9. Plaque Forming Experiment

Exponentially growing LMH cells were seeded on 6-well plates. One day later, when cells reached 90% confluence, diluted viruses in 1.5 mL DMEM plus 2% FBS was added to each well. After two hours’ inoculation, viruses were removed, and the cells were washed twice with 10 mM phosphate buffered saline (PBS) and covered with 2.5 mL DMEM containing 2% FBS and 0.7% low-melting agarose. Five days after infection, 2 mL fresh DMEM containing 2% FBS was supplemented to each well. At day 7 after infection, the liquid medium on the top was removed without damaging the semisolid overlay, and 2.5 mL 4% paraformaldehyde in PBS was added to each well. The cells were fixed at room temperature for 2 h, the semisolid medium was discarded, and cells were rinsed with running water before covering with 1.5 mL crystal violet solution for half an hour. After removing the staining solution, wells were rinsed with running water for several times and photographed [17]. The number of the plaques in each well was counted, and the area of each plaque was measured by using Fiji image processing package (http://fiji.sc/). The sizes of the plaques formed by different recombinant FAdV-4 were compared with the Kruskal–Wallis nonparametric test (GraphPad Prism 6).

### 2.10. Viral Inoculation of Embryonated Chicken Eggs

Specific pathogen free (SPF) chicken eggs were purchased from Beijing Boehringer Ingelheim Vital Biotechnology Company. Six day-old eggs were divided into four groups (15 eggs per group), inoculated with purified virus of 2 × 10^8^ vp in 100 μLPBS or only PBS via the yolk sac route, and cultured in a 37 °C incubator. The viability of the embryos was checked every 24 h. Livers from dead embryos were collected, weighed, smashed into small pieces, suspended in PBS and frozen in −80 °C. After three cycles of freeze-and-thaw, the liver suspension (six samples were selected from each virus-infected group) was spun at 1200× *g* for 5 min, and the supernatant was titrated on LMH cells with the limiting dilution method by counting fluorescence-positive cells. The yield of virus was normalized by the weight of the liver and presented as infectious units per gram of liver (IU/g). The study end point was set to the embryo age of 18 days, after that the viable embryos were killed by chilling the eggs at 4 °C overnight [18,19,20]. The data of embryonic lethality were subjected into survival analysis (GraphPad Prism 6).

## 3. Results

### 3.1. Reverse Genetics System for Modification of the Right End of FAdV-4 Genome

Schematic diagram of the establishment and application of the reverse genetics system was shown in Figure 1 and Appendix A. The fragment, which contained the right end of FAdV-4 genome, plasmid backbone and part of the left end of the viral genome, was separated from pKFAV4 by AvrII digestion and fused with artificially synthesized short DNA fragment, which contained the two overlaps for small plasmid assembly (OL-S1 and OL-S2), the two overlaps for large plasmid assembly (OL-L1 and OL-L2) and a PacI site, to generate the intermediate plasmid pKFAV4AP by DNA assembly. pKFAV4AP and pKFAV4 composed the reverse genetics system: the viral genes could be modified conveniently in pKFAV4AP, and the modified pKFAV4AP could be digested by PacI and restored to pKFAV4 by DNA assembly (Appendix A).

### 3.2. Construction of Recombinant FAdV-4 with Deletions of ORF1, ORF1b and ORF2

DNA assembly was employed to fuse three fragments together to form a small plasmid pKFAV4APN-GFP, which contained the same sequence of the short fragment of NheI-digested pKFAV4AP except that the CDS of ORF1, ORF1b and ORF2 was replaced with the GFP expression cassette (Figure 2). pKFAV4APN-GFP was restored to pKFAV4AP by restriction-ligation, and the generated plasmid was further restored to pKFAV4 by DNA assembly to produce the adenoviral plasmid pKFAV4-GFP. GFP focuses appeared on LMH cells three days after being transfected with PmeI-linearized pKFAV4-GFP. These focuses grew and merged, and complete cytopathic effect (CPE) finally occurred five days after transfection, suggesting successful rescue of recombinant virus FAdV4-GFP (Figure 3A). The genomic DNA of FAdV4-GFP was extracted and identified by restriction analysis and PCR (Figure 3B,C). The results indicated that FAdV4-GFP was successfully constructed.

### 3.3. Construction of Recombinant FAdV-4 with Deletions of ORF1, ORF1b, ORF2 and ORF4

ORF4 is a viral gene located at the very right end of the genome. The CDS of ORF4 in pKFAV4APN-GFP was replaced with that of mCherry through overlap extension PCR-mediated site-directed mutation (Figure 2). The generated pKFAV4APN-GX4C (GFP-xORF4-mCherry) plasmid, in which ORF1, ORF1b, ORF2 and ORF4 were deleted, was restored to pKFAV4 to form pKFAV4-GX4C with the same procedure of constructing pKFAV4-GFP. Recombinant FAdV4-GX4C was rescued similarly. Because it carried both GFP and mCherry genes, the focuses showed green and red fluorescence when observed under fluorescence microscope (Figure 4A). The genomic DNA was further identified by restriction analysis and PCR (Figure 4B,C). The results demonstrated that FAdV4-GX4C was correctly generated.

### 3.4. Modification of the Reverse Genetics Systems

On the map of the intermediate plasmid pKFAV4AP, it could be learned that it contained unique restriction sites of ApaI, PacI, HindIII, AscI, XbaI, BbvCI, SpeI and AbsI (Figure 1). Therefore, gene sequence among these restriction sites could be easily modified through one step of overlap extension PCR-mediated site-directed mutation. However, it spans 7 kb from AbsI to ApaI sites in the clockwise direction. There is no typical unique restriction site that can be used in the middle of this region, which makes modification of genes in this region more difficult. That is why a smaller plasmid pKFAV4APN-GFP was constructed and two steps of restoration were needed for generation of pKFAV4-GFP and pKFAV4-GX4C. To simplify the procedure, The AvrII site inside the fiber-2 gene was removed without change of the amino acid it encoded to generate a mutant pKFAV4 (pKFAV4M; Appendix A); and a smaller intermediate plasmid pKFAV4SAP-GFP was also constructed (Appendix A). pKFAV4SAP-GFP and pKFAV4M composed the modified reverse genetics system, which could be used to modify the regions upstream of the AvrII site at the left and downstream of the SpeI site at the right in the FAdV-4 genome (Appendix A).

### 3.5. Construction of Recombinant FAdV-4 with Deletion of ORF1, ORF1b, ORF2 and ORF19A

The modified system was validated by deletion of ORF19A. Overlap extension PCR-mediated site-directed mutations were carried out to delete the ORF19A CDS and replace the GFP with mCherry in pKFAV4SAP-GFP (Appendix A). The generated plasmid pKFAV7087-Che was linearized by PacI digestion and fused with the large fragment of SpeI/AvrII digested pKFAV4M to generate pKFAV4-CX19A (mCherry–xORF19A, Appendix A). Recombinant virus was rescued from LMH cells transfected with PmeI-linearized pKFAV4-CX19A. The viral genome (FAdV4-CX19A) was further identified by restriction analysis and PCR (Figure 5A,B). The results indicated that FAdV4-CX19A was correctly constructed.

### 3.6. Growth Property of Recombinant Viruses in LMH Cells

The purified recombinant FAdV-4 viruses had a physical particle-to-infectious unit (IU) ratio of 200 to 300. LMH cells in 12-well plate were infected with purified viruses at a multiplicity of infection (MOI) of 50 vp/cell for 2 h, and the progeny viruses were harvested 48 or 72 hafter infection and titrated on LMH cells. It could be seen that the FAdV4-CX19A had the highest replication ability, followed by FAdV4-GFP and FAdV4-GX4C in sequence (Figure 6A). The difference in viral replication in vitro was confirmed by plaque forming assay (Figure 6B). FAdV4-CX19A formed the largest plaques on LMH confluent monolayer cells while FAdV4-GX4C formed the smallest (Figure 6B,C). There was no significant difference in plaque numbers between the three recombinant viruses when the same amount viruses were used to infect LMH cells on six-well plates. One-step growth curve assay was further conducted to compare growth property of FAdV-4 and recombinant viruses (Appendix A). While FAdV-4 and FAdV4-GFP had similar growth characteristic in LMH cells, the replication rates successively increased among FAdV4-GX4C, FAdV4-GFP and FAdV4-CX19A. The results indicated that deletion of ORF4 or ORF19A altered the replication ability of FAdV-4.

### 3.7. Expression of mCherry Controlled by the FAdV-4 ORF4 Promoter in LMH Cells

In order to evaluate the possibility of expressing exogenous gene with FAdV-4 ORF4 original promoter, the coding sequence of ORF4 was replaced with that of mCherry reporter gene in FAdV4-GFP to generate FAdV4-GX4C. The expression of GFP or mCherry under CMV promoter in FAdV4-GX4C or FAdV4-CX19A served as controls. No expression of mCherry could be observed in FAdV4-GX4C-infected LMH cells 9 hafter infection while the expression of GFP could be seen clearly in the same culture. The fluorescence of mCherry became obvious in FAdV4-GX4C-infected LMH cells 24 hafter infection although it was still substantially weaker than GFP in the same cells or mCherry in FAdV4-CX19A-infected cells. At 48 hafter infection, the expression of mCherryincreased slightly in FAdV4-GX4C-infected LMH when the viruses started to lyse the cells and the released GFP in FAdV4-GX4C-infected and mCherry in FAdV4-CX19A-infected cultures could be seen in the media (Figure 7). The results suggested that ORF4 was a non-essential gene and the promoter of ORF4 could be employed to express exogenous gene although it was weak. 

### 3.8. Embryonic Lethality of Recombinant FAdV-4 Infection in Chicken

Embryonated eggs could be used as an experimental alternative to chickens to preliminarily evaluate the mortality of virus infection. The virulence of FAdV4-GFP, -GX4C and -CX19A was compared by inoculating embryonated chicken eggs via yolk sac route. The death of embryos inoculated with FAdV4-GX4C started from day 5 and ended at day 10 after infection with a mortality of 100%. FAdV4-GFP inoculation caused the embryos to die from day 8 after infection, and all infected embryos died within two days. The survival curve of FAdV4-CX19A-infected embryos shifted rightward by nearly one day when compared with that of FAdV4-GFP inoculation (Figure 8A). The results of survival analysis showed that the survival curve of FAdV4-CX19A was significantly different from that of FAdV4-GFP while the difference between FAdV4-GX4C and FAdV4-GFP was not statistically significant. All viruses could be recovered from the livers of dead embryos. It seemed that there was an enhanced replication of FAdV4-CX19A in chicken embryos when compared with that of the other two viruses although the difference was not significant (Figure 8B). Embryonated eggs were used to propagate FAdV-4 in the laboratory previously, and it could be deduced that FAdV4-GFPhad similar embryonic lethality with FAdV-4 [13].The results suggested that ORF19A was related to the virulence of FAdV-4 although it was nonessential for virus replication. Because the immune system of chicken embryo is in developmentand is different from that of newly hatched or adult birds [21], the effect of ORF19A deletion on FAdV-4 virulence in chickens deserves further study.

## 4. Discussion

The genus-specific genes of FAdV have no similarity to those of HAdV. The function of FAdV genus-specific genes has not been investigated, except for ORF1 (dUTPase), ORF8 (Gam-1) and ORF22 [22,23]. It has priorities to study the functions of FAdV genus-specific genes. These genes are engaged in the interaction between FAdV and the host cell or host immune system [2,8]. Elucidation of the gene functions will facilitate the development of attenuated vaccines, which havethe advantages over inactivated or subunit vaccines of high efficacy, easy inoculation and low costs [24,25,26,27]. On the other hand, gene function study will help the construction of FAdV-based gene delivery tools. FAdV vectors can well complement the use of those based on mammalian adenoviruses. For instance, FAdV vectors can be used to construct vector vaccines against other infectious diseases of fowl, or to take the place of HAdV in human gene therapies in situation the effect of HAdV vectors will be compromised because of the neutralizing antibodies. Other possible advantages of FAdV vectors include higher cloning capacity, more stable physicochemical properties and lower cost in vector production [28]. A user-friendly reverse genetics system may help expedite FAdV gene function study.

The FAdV vectors were constructed with the strategy of bacterial homologous recombination two decades ago [28]. This platform was further modified and improved by Eva Nagy and her colleagues through introducing phage recombinase [10]. Recently, CRISPR/Cas9 was employed as a tool to introduce double-stranded breaks at targeting site in FAdV genome to increase the percentage of positive recombinants in eukaryotic cells [29]. The homologous recombination in bacteria with phage recombinase and counter-selection technique were combined and used for modifying the FAdV-4 genome [30]. All these methods utilized the activity of recombinase inside cells.

Instead of eukaryotic or prokaryotic systems of homologous recombination, we employed the strategy of combining restriction enzyme digestion and isothermal DNA assembly for site-directed modification of FAdV-4 here, which were absolutely cell-free reaction systems [31,32]. The schematic diagram of our strategy, which we call restriction-assembly for convenience’s sake, was shown in Appendix A. A short fragment, in which the region for future modification locates, is excised from the plasmid of infectious clone by a dual cutter restriction enzyme; an intermediate plasmid is generated through two-fragment Gibson assembly; site-directed mutation is carried out in the intermediate plasmid with traditional methods such as overlap extension PCR; and finally the modified intermediate plasmid is linearized by restriction digestion and restored to the adenoviral genome through two-fragment DNA assembly. The approximately 100-bp long DNA fragment, which includes sequences flanking the dual cutter sites in the adenoviral genome and one extra restriction enzyme site for the future linearization of the modified intermediate plasmid, does the job of the left and right homolog arms in homologous recombination and mediates the generation of intermediate plasmid and future modified adenoviral plasmid. The restriction-assembly strategy is different from homologous recombination strategy in following aspects [33]. The presence of recombinase and identical sequences triggers homologous recombination, which could also be a cause for the instability of plasmids in which identical or repeat regions exist. In contrast, DNA exonuclease, polymerase and ligase are included in Gibson assembly [31], which ensures the assembly reactions proceed only between the short overlaps at the ends of but not inside DNA fragments; and recombinase-negative bacteria are generally chosen as the host cells for following plasmid transformation and propagation, which is good for keeping plasmids stable. Homologous recombination requires longer homolog arms than DNA assembly. Homolog arms about 50-bp in length are needed for bacterophage recombinase-mediated recombination, while efficient recombination in *Escherichia coli* BJ5183 strain demands identical sequences as long as hundreds of base pairs. DNA assembly can take place between overlaps as short as 15 bp if the value of the melting temperate (Tm) is more than 48 °C. In addition, generic materials such as restriction enzymes, chemically competent *Escherichia coli* cells, PCR reagents and DNA assembly kit can satisfy the whole procedure of restriction-assembly in labs of molecular biology. For the homologous recombination strategy, bacterial strains expressing recombinase and electroporation are often indispensable [34]. All these facts claim the advantages of restriction-assembly strategy.

The whole genomes of FAdV were thoroughly analyzed with methods of biological informatics although only three genus-specific genes have been studied so far [2,8,23]. It was reported that ORFs 0, 1, 1A,1B, 1C and 2 at the left and TR-2, ORFs 11, 17 and 19 at the right of the genome were nonessential genes for FAdV-9, and ORFs 16 and 17 were dispensable for FAdV-4 [10,22]. Here, we found that ORF1, ORF1b and ORF2 at the left end of the genome were nonessential for FAdV-4 replication. FAdV-C, in which species FAdV-4 and -10 are included, has species-specific genes of ORF4 and ORF19A at the rightmost end of the genome. ORF4 and ORF19A can only be found in FAdV-C. Therefore, it is conceivable that they might be responsible for HHS caused by FAdV-4. However, whether these species-specific genes contribute to the virulence of the novel FAdV-4 remains unknown.

FAdV-4 ORF4 is a US22 protein [2]. US22 family was first described in human cytomegalovirus (HCMV), members of which are clustered at the end of HCMV genome [35,36]. US22 proteins are found in all betaherpesviruses and some animal viruses such as gallidalphaherpesvirus, fowlpox virus and iridovirus [37]. US22 family is predicted to counter anti-viral responses by interacting with specific host proteins although little is known about the detailed functions [37,38]. Our results showed that the ORF4 was non-essential for viral replication in LMH cells or in chicken embryos. However, it played roles in viral propagation since ORF4-deleted virus formed smaller plaques (Figure 6). Deletion of ORF4 could not prolong the survival of virus-infected chicken embryos. The CDS replacement experiments suggested that the promoter of ORF4 might be a weak one in LMH cells considering low level expression of mCherry in FAdV4-GX4C infected LMH cells (Figure 7). 

FAdV ORF19 encodes a polypeptide with predicted lipase function. Most FAdVs have one ORF19 gene while FAdV-C contains a duplicate homolog ORF19A besides ORF19. To make things more intriguing, the highly pathogenic FAdV-4 loses ORF19 due to a 1966 bp deletion. However, it has been reported that the deletion of ORF19 does not contribute to the viral virulence [29,30]. We are interested in whether the remaining unique ORF19A influences the replication of the novel FAdV-4. Deletion of ORF19A enhanced viral propagation in LMH cells (Figure 6). A slight increase of viral yield in liver could also be observed in FAdV4-CX19A-infected chicken embryos. However, deletion of ORF19A in FAdV-4 made the embryos survive one more day when compared with those infected by FAdV4-GFP, an ORF19A-intact control (Figure 8), suggesting ORF19A participates in FAdV-4 pathogenesis. Whether and how the deletion of ORF19A reduces FAdV-4 virulence in chicken deserves further study, which may provide clues for developing attenuated vaccines against the novel FAdV-4 infection. 

In summary, we established a reverse genetics system to facilitate modification of the right end of the highly pathogenic FAdV-4 genome. We found the species-specific genes of ORF4 and ORF19A were dispensable for the growth of FAdV-4 although they influenced viral replication or virulence. This work will benefit the study of FAdV-4 gene function and vaccine development.

## Figures and Tables

**Figure 1 viruses-12-00301-f001:**
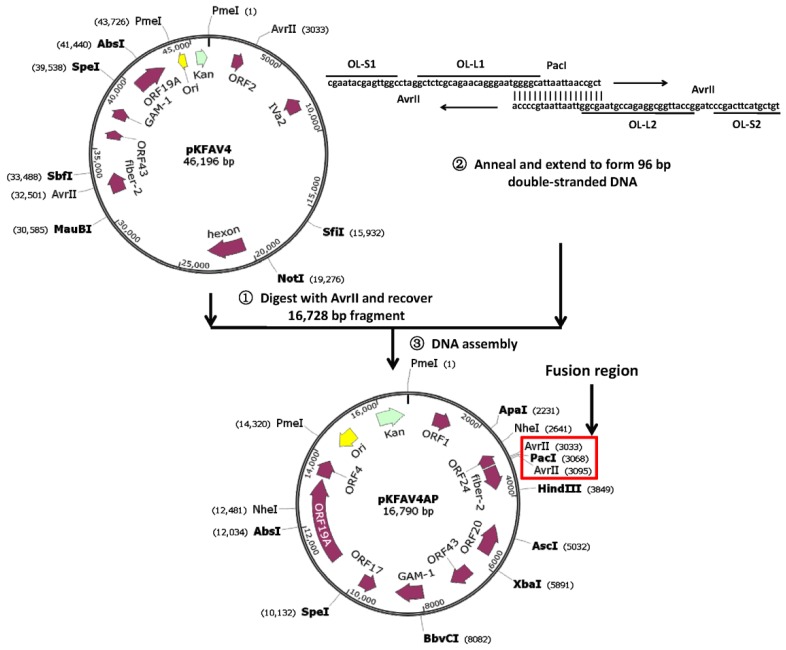
Schematic diagram for generating the intermediate plasmid pKFAV4AP (pKFAV4AvrII-PacI). The infectious clone pKFAV4 and pKFAV4AP composed a reverse genetics system for modification of the right end of fowl adenovirus 4 (FAdV-4): the gene mutation could be carried out in the small plasmid pKFAV4AP, and the modified gene could be restored to the viral genome by DNA assembly. The unique restriction sites were labeled in bold, which could be used for gene modification through overlap extension PCR combined with restriction-ligation cloning. The fusion region, including the restriction sites, were highlighted in the red box. OL-S1 and OL-S2: overlaps for generating the small plasmid (intermediate plasmid) by DNA assembly; OL-L1 and OL-L2: overlaps for generating large plasmid (adenoviral plasmid) by DNA assembly.

**Figure 2 viruses-12-00301-f002:**
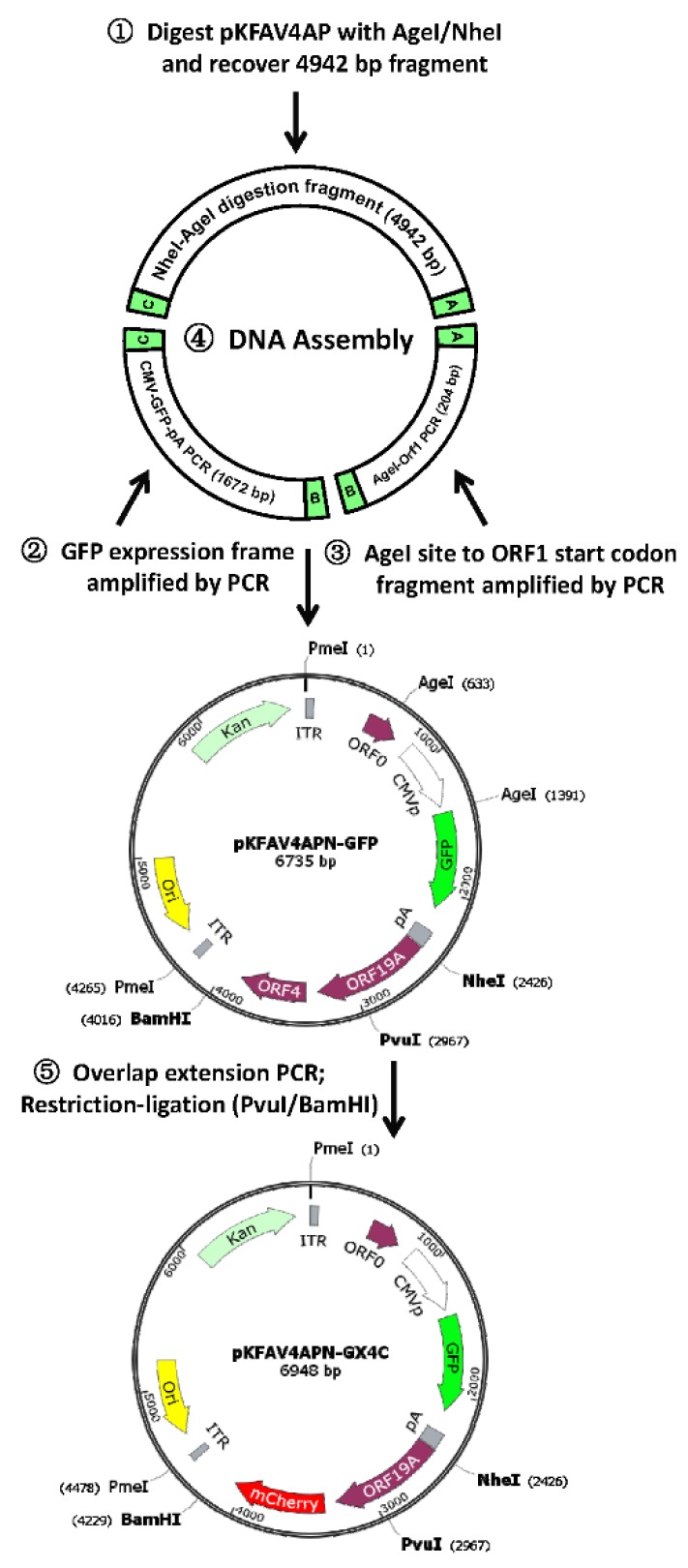
Schematic diagram for constructing small plasmid bearing fluorescent protein reporter genes. The DNA assembly of two PCR products and AgeI/NheI fragment in pKFAV4AP created plasmid pKFAV4APN-GFP (pKFAV4AP NheI-GFP), in which ORF1, ORF1b and ORF2 were replaced with Enhanced Green Fluorescent Protein (GFP) expression cassette. The ORF4 coding sequence (CDS) was further replaced with that of mCherry to generate pKFAV4APN-GX4C (GFP-xORF4-mCherry). These two small plasmids were separately restored to pKFAV4AP by restriction-ligation cloning.

**Figure 3 viruses-12-00301-f003:**
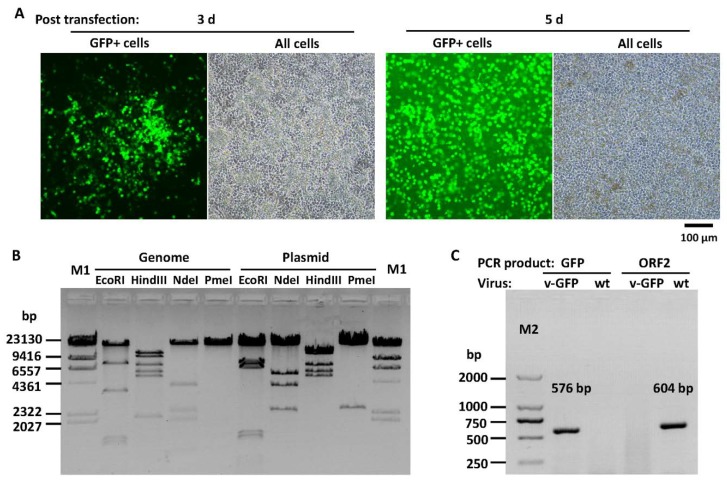
FAdV4-GFP rescue and identification. (**A**) Rescue of recombinant fowl adenovirus FAdV4-GFP. PmeI-linearized pKFAV4-GFP was used to transfect chicken hepatoma cells (Leghorn Male Hepatoma, LMH), and focuses formed by GFP-positive cells could be observed under a fluorescence microscope three days after transfection. Growth and fusion of these focuses led to cytopathic effect (CPE) five days after transfection. (**B**) Identification of FAdV4-GFP by restriction analysis of its genomic DNA with adenoviral plasmid pKFAV4-GFP as the control. The predicted molecular weights of digested fragments of the FAdV4-GFP genome were 463, 516, 699, 1248, 1384, 3584, 7836 and 27,774 bp for EcoRI; 2151, 5216, 5900, 7413, 10,468 and 12,356 bp for HindIII; 251, 1016, 1975, 2352, 3993 and 33,917 bp for NdeI; and PmeI did not cut. The predicted molecular weights of digested fragments of pKFAV4-GFP plasmid were 463, 699, 1248, 1384, 6577, 7836 and 27,774 bp for EcoRI; 5216, 5900, 7413, 12,356 and 15,096 bp for HindIII; 251, 2352, 3993, 5468 and 33,917 bp for NdeI; and 2471 and 43,510 for PmeI. (**C**) Identification of FAdV4-GFP by PCR. PCR was performed to amplify fragments inside GFP or ORF2 genes using the FAdV4-GFP genomic DNA (v-GFP) as the template, and the products were resolved on 1% agarose gel. Wild-type FAdV-4 genome (wt) served as a control template. The results indicated that FAdV4-GFP was successfully constructed.

**Figure 4 viruses-12-00301-f004:**
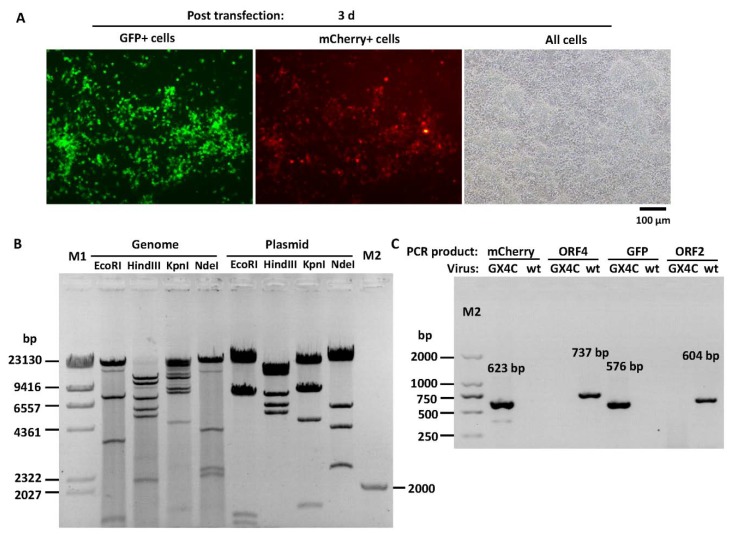
FAdV4-GX4C rescue and identification. (**A**) Rescue of recombinant fowl adenovirus FAdV4-GX4C. PmeI-linearized pKFAV4-GX4C was used to transfect LMH cells, and focuses formed by GFP- and mCherry-positive cells could be observed under a fluorescence microscope three days after transfection. (**B**) Identification of FAdV4-GX4C by restriction analysis of its genomic DNA with adenoviral plasmid pKFAV4-GX4Cas the control. The predicted molecular weights of digested fragments of the FAdV4-GX4C genome were 463, 1248, 1384, 1428, 3584, 7836 and 27,774 bp for EcoRI; 2151, 5216, 5900, 7413, 10,681 and 12,356 bp for HindIII; 935, 1542, 4546, 7990, 8123, 8840 and 11,741 bp for KpnI; and 251, 1016, 2188, 2352, 3993 and 33,917 bp for NdeI. The predicted molecular weights of digested fragments of pKFAV4-GX4C plasmid were463, 1248, 1384, 7489, 7836 and 27,774 bp for EcoRI; 5216, 5900, 7413, 12,356 and 15,309 bp for HindIII; 935, 1542, 4546, 7990, 8123 and 23,058 bp for KpnI; and 251, 2352, 3993, 5681 and 33,917 bp for NdeI. (**C**) Identification of FAdV4-GX4C by PCR. PCR was performed to amplify fragments inside mCherry, ORF4, GFP or ORF2 genes using the FAdV4-GX4C genomic DNA (GX4C) as the template, and the products were resolved on 1% agarose gel. Wild-type FAdV-4 genome (wt) served as a control template.

**Figure 5 viruses-12-00301-f005:**
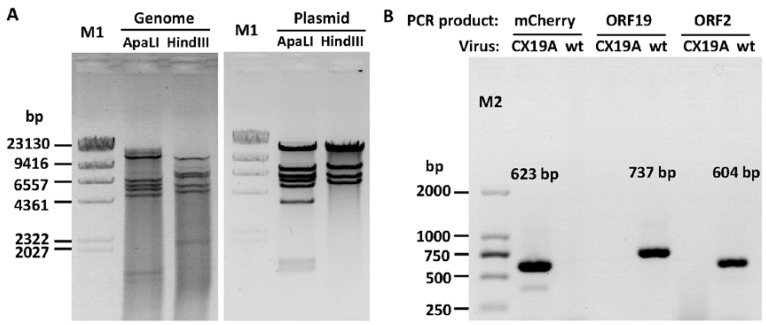
Identification of FAdV4-CX19A virus. (**A**) FAdV4-CX19A (mCherry–xORF19A) was identified by restriction analysis of its genomic DNA, and adenoviral plasmid pKFAV4-CX19A served as the control. The predicted molecular weights of digested fragments of the FAdV4-CX19A genome were 1312, 1447, 1450, 4879, 5592, 6113, 6904 and 13,354 bp for ApaLI; and 2119, 5216, 5900, 7413, 8047 and 12,356 bp for HindIII. The predicted molecular weights of digested fragments of pKFAV4-CX19A plasmid 1312, 1450, 3527, 4879, 5592, 6113, 7301 and 13,354 bp for ApaLI; and 5216, 5900, 7413, 12,356 and 12,643 bp for HindIII. (**B**) Identification of FAdV4-CX19A by PCR. PCR was performed to amplify fragments inside mCherry, ORF19A or ORF2 genes using the FAdV4-CX19A genomic DNA (CX19A) as the template, and the products were resolved on 1% agarose gel. Wild-type FAdV-4 genome (wt) served as the control template.

**Figure 6 viruses-12-00301-f006:**
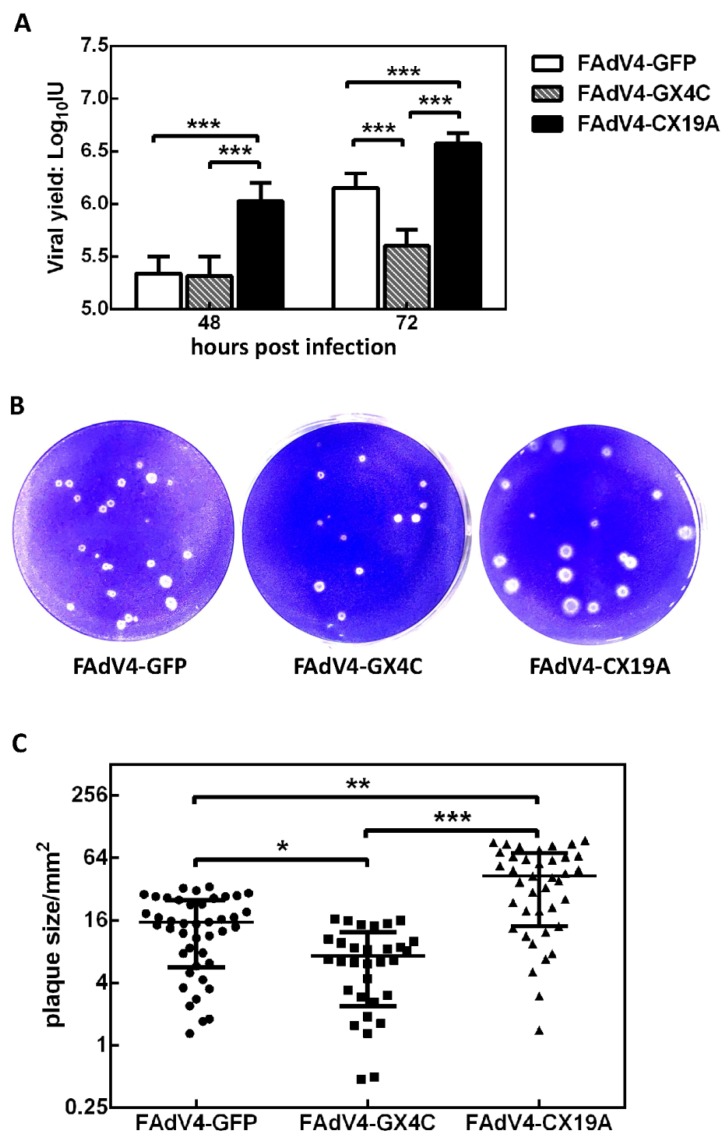
Viral replication in LMH cells. (**A**) LMH cells in 12-well plates were infected with FAdV4-GFP, FAdV4-GX4C or FAdV4-CX19A at a multiplicity of infection (MOI) of 50 vp/cell for 2 h. The progeny viruses were harvested 48 or 72 h after infection, and titrated on LMH cells. The viral yield (infectious units, IU) in one well was calculated and shown. (**B**) LMH cells in 6-well plates were infected with FAdV4-GFP, FAdV4-GX4C or FAdV4-CX19A at an amount of 8000 vp/well for 2 h. Cells were covered with semi-solid culture medium and cultivated for seven days before fixing and staining to show the plaques. (**C**) The area of each plaque was measured with the help of Fiji image processing package. The data from the three viruses were collected and analyzed with the Kruskal–Wallis nonparametric test. * *p* < 0.05, ** *p* < 0.01, and *** *p* < 0.001.

**Figure 7 viruses-12-00301-f007:**
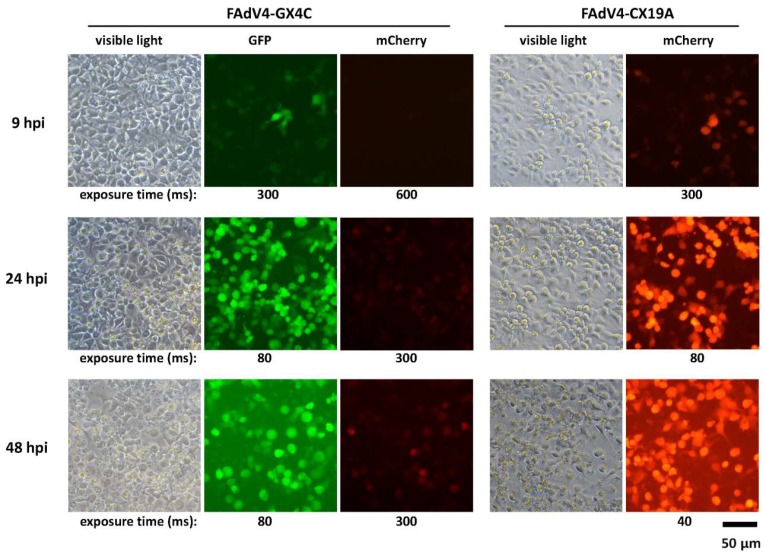
Expression of mCherry under the ORF4 promoter of FAdV-4. LMH cells were infected with FAdV4-GX4C and FAdV-CX19A. The expression of GFP and mCherry was observed under fluorescence microscope at 9, 24 and 48 hafter infection (hpi). In FAdV4-GX4C, GFP was controlled by the CMV promoter while mCherry was controlled by the ORF4 original promoter. In FAdV4-CX19A, mCherry was controlled by the CMV promoter. The results indicated that the ORF4 original promoter was a weak one in LMH cells.

**Figure 8 viruses-12-00301-f008:**
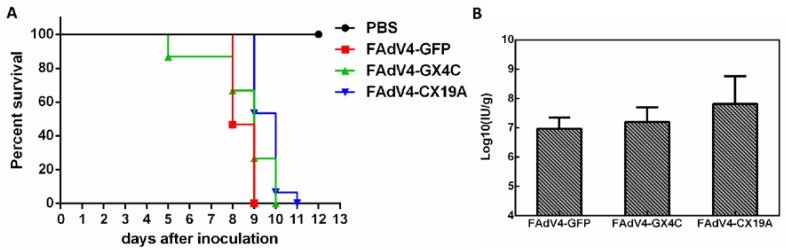
Inoculation of embryonated chicken eggs with recombinant FAdV-4. Sixty six-day old embryonated chicken eggs were randomly divided into four groups (15 eggs per group), and three groups were inoculated with FAdV4-GFP, FAdV4-GX4C or FAdV4-CX19A of 2 × 10^8^ vp in 100 μLPBS via yolk sac route, respectively. The PBS group was injected with 100 μLPBS containing no virus, and served as uninfected control. The viability of embryos was checked every 24 h. Livers from dead embryos were weighed and collected for virus titration. (**A**) Survival curve of embryonated chicken eggs after viral inoculation. (**B**) Virus amount in liver normalized by liver weight, which was shown as infectious units per gram of liver (IU/g).

**Table 1 viruses-12-00301-t001:** Summary information of oligonucleotides.

Fragment	Oligo Name	Sequence	Template	Product Length (bp)
AvrII-PacI	1707Avr-PacF	cgaatacgagttggcctaggctctcgcagaacagggaatggggcattaattaaccgct	self-anneal	96
1707Avr-PacR	tgtcgtacttcagccctaggccattggcggagaccgtaagcggttaattaatgcccca
AgeI-ORF1	1707KFAV4AgeIF	attcctccactgctttgaaccca	pKFAV4AP	204
1707KFAV4AgeIR	cccgtaattgattactattaccttgtagaaaaagagagaaaattg
GFP cassette	1707F02GFPF	ttctacaaggtaatagtaatcaattacggggtcattagtt	pAd5GFP	1672
1707F02GFPR	tcgatttactgtgaagctacaagtgctagctaagatacattgatgagtttggacaaac
PvuI-ORF4	1711FAV4GCX5	ggaaccgatcgaagaaagcaacag	pKFAV4-GFP	424
1711FAV4GCX6	gcccttgctcaccatgtcagaatatatagagaaaggaatgggc
mCherry CDS	1711FAV4GCX7	ctctatatattctgacatggtgagcaagggcgaggag	pmCherry-N1	742
1711FAV4GCX8	ctggaatatagtgtgttacttgtacagctcgtccatgccg
ORF4-BamHI	1711FAV4GCX9	ggacgagctgtacaagtaacacactatattccagtccgaggagg	pKFAV4-GFP	180
1711FAV4GCX10	taagtggatccgcacaccattgc	1281
117-bp linker	1812FAV4APf	cgaaccagtaggcgaatacgagttggcct	pKFAV4AP	117
1812FAV4APxAr	gggtattagtgtcgtacttcagcccAaggccattggcggagacc
116-bp linker	1812FAV4SAP1	ccagtaggcgaatacgagttggcctaggctctcgcagaacagg	self-anneal	116
1812FAV4SAP2	tatctctatgctttgTTAATTAAcccattccctgttctgcgagagcctag g
1812FAV4SAP3	aatgggTTAATTAAcaaagcatagagataaaagaaacccgttactagtccagga
1812FAV4SAP4	ggatttaggaaaagtgtttcctggactagtaacgggtttctt
SpeI-BstZ17I	1902FAdV4Xorf19a1	caaagcatagagataaaagaaacccgttac ta	pKFAV4AP	804
1902FAdV4Xorf19a2	tacaacatttcagtagtttcctggtatacattgatgtgaccttcatggcgaac
CMVp	1904FAV4MCHE1	tcaagtgtatcatatgccaagtacg	pKFAV4-GFP	395
1904FAV4MCHE2	ccttgctcaccatggtaggcctctagcggatctgacggttcact
mCherry CDS	1904FAV4MCHE3	ctagaggcctaccatggtgagcaagggcgagga	pmCherry-N1	733
1904FAV4MCHE4	ggccgtcgactacttgtacagctcgtccatgc	1102
mCherry	1906mCHEF1	ggccatcatcaaggagttcatg		623
1906mCHER1	gttccacgatggtgtagtcctcg	
GFP	1906GFPf	cggccacaagttcagcgtgt c		576
1906GFPr	cgcttctcgttggggtcttt g	
ORF2	1905ORF2f	ggcttcggaccgttactggg		604
1905ORF2r	gggggtacggttaatctccc	
ORF19A/ORF4	1905ORF19Af	ctcccgttcaaagtagtgaa ca		737
1906ORF4r	ggcagatacagcacttcgcagta

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
