# Peer review of "User-Friendly Reverse Genetics System for Modification of the Right End of Fowl Adenovirus 4 Genome"

_viruses, 2020, doi:10.3390/v12030301_

Round 1

Reviewer 1 Report

The authors described an easy-to-use reverse genetics system for modification of the right end of the novel FAdV-4 genome. The work is a supplement to previous reported FAdV-4 reverse genetic system.  However there are several drawbacks needs to be improved.

  1. The title of the manuscript states that the system is for modification of the right end of FAdV-4 genome. However, as we all know, the ORF1, 1b and 2 located at the left side of FAdV-4 genome. This will confuse the readers. The focus of the title is different from the content.
  2. In the Abstract and Introduction section, the authors should state a clear objective of the study. There was a lack of explanation why the previous reverse genetics systems are not user-friendly, even the authors give brief statement in the Discussion section. The authors should know that every method has its own strengths and weaknesses.
  3. Improvements in figure presentation could help readers to understand better. In Method 2.3 and Result 3.2, pKFAV4AP was digested with Agel/Nhel to get a 4942 bp sequence, but AgeI restriction digestion site was not shown on pKFAV4AP plasmid according to Figure 1. The labels on the plasmids were not consistent to each other which could be confusing to readers who are not familiar with the subjects. For example, in Figure 1, more and different ORFs were labelled out in pKFAV4AP comparing to pKFAV4 at the same regions of the two plasmids.

  4. More technique clarifications should be provided in method sections to assist repetitions of the experiment, such as bacteria strain for transformation and the condition of transformation.

  5. The designed names of the DNA constructs are too confused to be understood, such as pKFAP, pKFAPN,FAV4GCX, pKFAV4-GX4C,pKFAV4APX, pKFAV4SAP-GFP, etc. Please simplify them.
  6. The titration method for recombinant virus is debatable. The authors determined the viruses particle titer by measuring the content of genomic DNA where 100 ng of genomic DNA is equivalent to 2.3×109 viral particles, since a 43-kb genome has a molecular mass of 2.6×107. This method won't give accurate results, and the particle does not reflect the infectious titer which the authors used for infection experiment. They should take more efforts to study fowl adenovirus-related literature.  As all the recombinants can produce CPE in LMH cells, I would recommend to carry out one-step growth curve assay.
  7. When the authors evaluate the growth property and embryonic lethality of the recombinants, the wild type FAdV-4 should be included for comparison.
  8. In the result of embryonic lethality experiment, the authors should describe and present other signs and lesions produced in the embryo, rather than the survival rate only. And it should be note that no correlation can be found between high embryo mortality and clinical signs in experimentally infected day‐old specific pathogen free birds. So the conclusion of ORF4 and ORF19A influence virulence is questionable

Author Response

Reviewer #1:

  1. The title of the manuscript states that the system is for modification of the right end of FAdV-4 genome. However, as we all know, the ORF1, 1b and 2 located at the left side of FAdV-4 genome. This will confuse the readers. The focus of the title is different from the content.

This system was designed for modification of the right end of FAdV-4 because more genes that we were interested in were located on this end. The intermediate plasmid contained the whole right end (downstream of the fiber gene), and any gene on this region could be modified with this system. Modification of the right end is the main function of this reverse genetics system, so we wish that we could keep this title. To minimize the confusion, we changed a sentence to: “We established an easy-to-use reverse genetics system for modification of whole right and partial left ends of the novel FAdV-4 genome” in the abstract (line 22).

  1. In the Abstract and Introduction section, the authors should state a clear objective of the study. There was a lack of explanation why the previous reverse genetics systems are not user-friendly, even the authors give brief statement in the Discussion section. The authors should know that every method has its own strengths and weaknesses.

We do agree with the statement that every method has its own strengths and weaknesses.

We deleted “However, user-friendly systems are still lacking ....”, and added: “Most of these systems work through homologous recombination, which needs the activity of intracellar recombinase” (line 64-65).

We deleted: “However, the study of FAdV gene functions is hampered by ....”, and added “A user-friendly reverse genetics system may help expedite FAdV gene function study” (line 409-410, Discussion section).

  1. Improvements in figure presentation could help readers to understand better. In Method 2.3 and Result 3.2, pKFAV4AP was digested with Agel/Nhel to get a 4942 bp sequence, but AgeI restriction digestion site was not shown on pKFAV4AP plasmid according to Figure 1. The labels on the plasmids were not consistent to each other which could be confusing to readers who are not familiar with the subjects. For example, in Figure 1, more and different ORFs were labelled out in pKFAV4AP comparing to pKFAV4 at the same regions of the two plasmids.

There are 4 AgeI sites in pKFAV4AP. If we label them on the map, it might distract readers from the unique restriction sites which could be used for modifying FAdV-4 right end. That is why we chose not to label AgeI sites on the pKFAV4AP map in Figure 1. The flaw could be slightly fixed in Figure 2 where NheI/AgeI sites were labelled on pKFAV4APN-GFP map. pKFAV4 plasmid contains the whole genome of FAdV-4, and it is very difficult to label all known genes on this map due to space restriction. pKFAV4AP contains partial genome (if we zoom in on the right end of pKFAV4 we get pKFAV4AP), and there is bigger space for labelling more right-end genes in pKFAV4AP map. To partially satisfy the professor’s requirement, we labelled two more genes on pKFAV4 map to coordinate these two plasmid maps.

  1. More technique clarifications should be provided in method sections to assist repetitions of the experiment, such as bacteria strain for transformation and the condition of transformation.

We added one short paragraph in 2.1. “Plasmid transformation was performed on Escherichia coli TOP10 chemically competent cells with the standard heat shock procedure according to the manufacturer’s instructions (TIANGEN Biotech, Beijing, China).” (line 82-84).

  1. The designed names of the DNA constructs are too confused to be understood, such as pKFAP, pKFAPN,FAV4GCX, pKFAV4-GX4C,pKFAV4APX, pKFAV4SAP-GFP, etc. Please simplify them.

To make them easier to understand, we explained the plasmid names in parentheses the first time they appeared in the main text and figure captions.

pKFAV4 (plasmid bearing Kanamycin-resistant gene and FAdV-4 genome)

pKFAV4AP (pKFAV4 AvrII-PacI),

pKFAV4APN-GFP (pKFAV4AP NheI-GFP),

pKFAV4APN-GX4C (pKFAV4APN  GFP-xORF4-mCherry),

pKFAV4APX (pKFAV4AP xAvrII),

pKFAV4M (pKFAV4 AvrII site-Mutated),

pKFAV4SAP-GFP (pKFAV4 SpeI-AvrII-PacI GFP),

pKFAV4SAPX19a-GFP (pKFAV4SAP xORF19A GFP)

pKFAV4-CX19A (pKFAV4 mCherry xORF19A)

We have to give names to these plasmids, and we hope this arrangement is appropriate.

  1. The titration method for recombinant virus is debatable. The authors determined the viruses particle titer by measuring the content of genomic DNA where 100 ng of genomic DNA is equivalent to 2.3×109 viral particles, since a 43-kb genome has a molecular mass of 2.6×107. This method won't give accurate results, and the particle does not reflect the infectious titer which the authors used for infection experiment. They should take more efforts to study fowl adenovirus-related literature. As all the recombinants can produce CPE in LMH cells, I would recommend to carry out one-step growth curve assay.

Several years ago, scientists liked to use plaque forming unit (PFU) to evaluate the biological activity. After more adenoviruses are studied, or when people want to compare the infectivity between different adenoviruses, PFU is not a fair gauge any more because different adenoviruses often have various particle-to-PFU ratios and different titers can be detected even for the same batch of adenovirus if different cell lines are used as the host cells for titration. In contrast, viral particle is an objective physical unit for every adenovirus and become popular recently. For FAdV, just as the professor mentioned, PFU or TCID50 are commonly used. We think there might be a reason behind that: FAdVs are usually used in the form of raw cellular lysate, scientists don’t routinely purify FAdVs and the viral particle unit is hard to be determined in this situation. In our manuscript, we purified the viruses and determined the viral particle titer as well as infectious unit (IU) titer. The particle-to-IU ratios were calculated to be 200 to 300 (3.6. line 323). We want to stick to the use of viral particle unit in our manuscript since it brings little inconvenience to the readers and is understandable if they know the particle-to-IU ratios.  

  1. When the authors evaluate the growth property and embryonic lethality of the recombinants, the wild type FAdV-4 should be included for comparison.

To follow the professor’s suggestion, we added one more supplementary figure of one-step growth curves of wild-type and recombinant FAdV-4 viruses (Figure 4S). There are some reasons that we didn’t include the wild-type FAdV-4 in this study. Firstly, we used wild-type FAdV-4 to infect embryonated chicken eggs to amplify virus for constructing infectious clones in our last publication, it was found that FAdV-4 infected embryos usually died in 7–10 days (mostly in 8-9 days), which is similar to FAdV4-GFP-infected ones [13]. Secondly, FAdV4-GFP and wild-type FAdV-4 have similar growth characteristic according to the one-step growth curves (Figure 4S). Thirdly, we aimed to observe the effects of ORF4 and ORF19A genes on the virus growth, and FAdV4-GFP could serve as the appropriate control since all of the three recombinant FAdV-4 viruses have the same background deletion of ORF1, ORF1b and ORF2. In contrast, wild-type FAdV-4 doesn’t have such deletions. If we aim to study the function of ORF1, ORF1b and ORF2, FAdV4-GFP will be the target gene-knockout model and FAdV-4 can be used as the perfect control. Finally, all of the recombinant viruses carry reporter genes, which simplified the experimental procedures. It is really inconvenient to include FAdV-4 in this study.

We added one sentence in 3.6 (line 331-334): “One-step growth curve assay was further conducted to compare growth property of FAdV-4 and recombinant viruses (Figure S4). While FAdV-4 and FAdV4-GFP had similar growth characteristic in LMH cells, the replication rates successively increased among FAdV4-GX4C, FAdV4-GFP and FAdV4-CX19A.”

We added another sentence in 3.8 (line 381-382) “Embryonated eggs were used to propagate FAdV-4 in the laboratory previously, and it could be deduced that FAdV4-GFP had similar embryonic lethality with FAdV-4 [13].”

[13] Zou, X.H.; Bi, Z.X.; Guo, X.J.; Zhang, Z.; Zhao, Y.; Wang, M.; Zhu, Y.L.; Jie, H.Y.; Yu, Y.; Hung, T., et al. DNA assembly technique simplifies the construction of infectious clone of fowl adenovirus. J Virol Methods 2018, 257, 85-92, doi:10.1016/j.jviromet.2018.04.001.

  1. In the result of embryonic lethality experiment, the authors should describe and present other signs and lesions produced in the embryo, rather than the survival rate only. And it should be note that no correlation can be found between high embryo mortality and clinical signs in experimentally infected dayold specific pathogen free birds. So the conclusion of ORF4 and ORF19A influence virulence is questionable.

Liver is the most important target organ for FAdV-4 infection. We conducted histochemistry experiments and observed the lesions on livers of dead embryos. There was no obvious difference between the virus-infected groups. This result was in agreement with the virus titration data (large amount of virus could be found in livers of all virus-infected embryos). Virus yield data were quantitative, and we think it might be appropriate to show this result only due to space limitation.

The immune system of chicken embryos is in development. It is not mature and cannot function as that of adult birds. However, chicken embryos do have immune system and immunocytes. Lethality of FAdV-4 on embryos can give some clues to the virus virulence on newly hatched or adult birds. “The results suggested that ORF19A was related to the virulence of FAdV-4 although it was non-essential for virus replication” is not a strong conclusion. To avoid misunderstanding, we added one more sentence at the end of the paragraph (3.8, line 384-386): “Because the immune system of chicken embryo is in development and is different from that of newly hatched or adult birds [21], the effect of ORF19A deletion on FAdV-4 virulence in chickens deserves further study.”

Reviewer 2 Report

This paper is well written and presented. A problem is that it is largely a methods paper. The authors use rather old-fashioned (but reliabble) techniques to make genomic constructs of FAdV, A merit of the paper is that their reagents coud be valuable for other invetigators in the field, if made available. A nunber of mutants were made with their technology. Unfortunately these showed very minor phenotypes like changes in plaque size or speed in killing infected animals. These observations do not allow any mechanistic interpretations.

It is a very ambitious paper and if there is space in the journal I would recommend publication. The text could be shortened as the constructions made are well described in the figures and the figure legends.

Author Response

The other reviewer professor suggested us to provide more details of the experiments. Since the length of our manuscript has not exceeded the word limit required by the journal, we added some concise explanations. Thanks.

Round 2

Reviewer 1 Report

Basically, the authors have answered all the comments. I think the manuscript is suitable for publication now.

Regarding the titration method for adenovirus, I do agree with the author that viral particle is an objective physical unit. Viral particle is usually used to evaluate the quality of a purified virus stock in adenovirus field. However when people determine the infectivity, you still have to consider the CPE. As genomic DNA can still be extracted from inactivated viruses, but the viral stock has lost its infectivity. Of course, the PFU method also has its weakness such as inconsistent results between different measuring time or different people performing the titration, etc.  It is normal that different titers can be detected even for the same batch of adenovirus if different cell lines are used as the host cells for titration. Because susceptibility of each cell line to a same virus could be different.